# Dietary Supplementation with *Agave tequilana* (Weber Var. Blue) Stem Powder Improves the Performance and Intestinal Integrity of Broiler Rabbits

**DOI:** 10.3390/ani12091117

**Published:** 2022-04-27

**Authors:** Yordan Martínez, Maidelys Iser, Manuel Valdivié, Manuel Rosales, Esther Albarrán, David Sánchez

**Affiliations:** 1Agricultural Science and Production Department, Zamorano University, Valle de Yeguare, San Antonio de Oriente, Francisco Morazan, P.O. Box 93, Tegucigalpa 11101, Honduras; 2Centro Universitario de Ciencias Biológicas y Agropecuarias (CUCBA), Universidad de Guadalajara, P.O. Box 49, Guadalajara 44214, Jalisco, Mexico; sofiaizabellamai@gmail.com (M.I.); manuel.rcortes@academicos.udg.mx (M.R.); esther.albarran@academicos.udg.mx (E.A.); david.schipres@academicos.udg.mx (D.S.); 3National Center for Laboratory Animal Production, Santiago de las Vegas, Rancho Boyeros, La Habana P.O. Box 6240, Cuba; mvaldivie@ica.co.cu

**Keywords:** agave, intestinal mucosa, natural products, rabbit

## Abstract

**Simple Summary:**

Natural products have been used as an alternative to the indiscriminate use of subtherapeutic antibiotics for improving the growth of animals. Although these synthetic products have been used in rabbits to a lesser extent than in other monogastrics, several natural products have been tested with positive responses in growth performance, immune activity, and antioxidant capacity. Agaves are plants with chemical compounds that have shown positive effects on performance and animal health. Specifically, the *Agave tequilana* stem has a high content of fructans and secondary metabolites, such as polyphenols and saponins, that are responsible for various biological activities. The dietary use of up to 1.5% of *Agave tequilana* stem powder in the diet of rabbits had a natural growth-promoting effect due to the improvement of their intestinal integrity (with an emphasis on the concentric layers, villi, and crypts), taken as indicator of intestinal health. Considering the results of this study and others carried out by this group of authors, this natural product (*A. tequilana*) could be utilized in rabbit farming, considering that the stem of the agave is currently not utilized for anything.

**Abstract:**

This study evaluated the effect of *Agave tequilana* (Weber var. azul) stem powder on the growth performance and the intestinal integrity in rabbits. A total of 120 male rabbits [New Zealand × California] were weaned for 35 days and randomized into four dietary treatments, 15 replicates per treatment, and two rabbits per replicate. The treatments consisted of a basal diet (T0) and dietary supplementation with 0.5% (T1), 1.0% (T2) and 1.5% (T3) of *Agave tequilana* stem powder. The T3 treatment improved the body weight and average daily gain (*p* < 0.05) compared to the other groups, without affecting viability and feed conversion ratio (*p* > 0.05). Furthermore, the T3 treatment enhanced (*p* < 0.05) the thickness of the muscular and mucous layers, and the height, thickness, and number of villi in the duodenum (*p* < 0.05). However, this treatment (T3) significantly decreased (*p* < 0.05) values for the area and depth of the crypts in the duodenum and the villus/crypt ratio. Likewise, in the cecum, T3 treatment provoked a marked decrease (*p* < 0.05) in the depth and thickness of the crypts. The results indicate that the dietary use with 1.5% of *A. tequilana* stem powder had a natural growth-promoting effect and enhanced the histomorphometry of the concentric layers (muscle and mucosa), villi, and crypts as indicators of intestinal health in rabbits.

## 1. Introduction

The constant consumption of synthetic antimicrobials causes the presence of chemical residues in products of animal origin, which directly affects human health [1]. Although preventive antibiotics have been used more in poultry and pig production than in rabbit production, some reports have focused on controlling the population of *Enterobacteriaceae* with bacitracin Zn [2]. Thus, organic acids, prebiotics, probiotics, and medicinal plants have been used efficiently as viable alternatives to replace or decrease the indiscriminate use of growth-promoting antibiotics [3].

In animals, studies are carried out with natural products to evaluate intestinal health indicators, which are important for justifying the growth-promoting effect and the bactericidal, anti-inflammatory and antioxidant activities of these foods—three key considerations for nutraceuticals [4]. Furthermore, intestinal integrity, pH, and gut microbiology are known to be specifically influenced by diet and gut health [5]. In this sense, dietary use of nutraceutical products such as prebiotics and probiotics can increase the height of the villi in the small intestine, without changing the crypts depth and the natural promotion of growth in farm animals [6,7,8,9].

Plants of the genus Agaves have been used by the inhabitants of Mesoamerica for 9000 years, especially in Mexico where it originated. Of the 310 reported species, 272 of them are endemic to Mexico [10]. Among the most recognized and economically important species is the *Agave tequilana* Weber var. azul [11]. Agaves are plants with a high content of fructans synthesized and stored in the stems, and made up of polymers of fructose derived from the sucrose molecule and with glucose as a terminal monomer (generally). Moreover, the structure of fructans is used as a taxonomic marker in agaves. Specifically, the stem of the *Agave tequilana* Weber var. azul has mainly β (2→1) fructooligosaccharide linkages; and some ramifications of β (2→6) are considered to be a very complex chemical structure whose quantification will depend on the plant material (*Agave* spp.) under study [12]. Due to the high levels of fructans in *Agave tequilana*, prebiotic nutraceutical products have been obtained for their dietary use in human beings and animals [13].

Studies on farm animals reported that *Agave tequilana* stem powder acted as a natural growth promoter in diets by increasing the population of beneficial cecal bacteria and by modifying harmful serum lipids in pigs and poultry [14,15]. Likewise, authors such as Iser et al. [16] and Iser et al. [17] have demonstrated that dietary supplementation of up to 1.5% *Agave tequilana* stem powder increased productivity and meat quality, decreased harmful serum lipids, and left blood indicators of fattening rabbits unchanged. However, the study of how *A. tequilana* stem powder influences intestinal histomorphology as indicators of intestinal health could justify the growth-promoting effect of this natural prebiotic in rabbit farming. Therefore, the objective of this experiment was to evaluate the performance and histo-morphometry of the concentric layers (muscle and mucosa), villi, and crypts of the duodenum and cecum in broiler rabbits fed with supplementations of *Agave tequilana* (Weber var. Azul) powder.

## 2. Materials and Methods

### 2.1. Experimental Location

The experiment with the name CINV.106/12 was approved by the Animal Care Committee of the Faculty of Veterinary Medicine and Zootechnics of the University Center for Biological and Agricultural Sciences, University of Guadalajara (CUCBA), Mexico. It should be noted that the Mexican animal welfare guidelines and the experimental protocol were followed. The animals were housed in an open shed, and temperature (21 °C ± 2) and relative humidity (63% ± 2) were measured daily using a hygro-thermometer (Jumbo Dig).

### 2.2. Animals, Treatments, Experimental Conditions, and Diets

From a total of 350 rabbits, 120 male rabbits (New Zealand × California) that had been weaned for 35 days were selected and randomized into 4treatments, 15 replicates per treatment, and 2 rabbits per cage. The experimental treatments consisted of a basal diet formulated according to the nutritional requirements of the species under study (T0) and dietary supplementation with 0.5% (T1), 1.0% (T2), and 1.5% (T3) of *Agave tequilana* stem powder. This natural product was offered by the Veterinary Division of the *Centro Universitario de Ciencias Biológicas y Agropecuarias* (CUCBA), University of Guadalajara, Mexico. The product has the following ingredients: 93.15% dry matter; 5.08% crude protein; 1.38% crude fat; 5.15% ash; 15.98% neutral detergent fiber; 7.70% acid detergent fiber; 8.28% cellulose; and 43.24% fructans (inulin type), according to AOAC 2001.11 [18], Van Soest et al. [19], and high-performance liquid chromatography (HPLC Varian ProStar System, Palo Alto, CA, USA). Supplementation levels from a previous experiment by Iser et al. [17] were also considered for this experiment. A basal diet was prepared according to the nutritional requirements reported by Blas and Mateos [20] for fattening rabbits (Table 1).

The *A. tequilana* stem powder was added homogeneously to the mash feed, and the mixture obtained was pelleted using a rotary pelletizer (Macreat, LDS-300, Zhengzhou, China) with a particle size of 2.5 mm as established for this animal species [20].

The broiler rabbits were placed in metal cages with dimensions of 76 cm long, 76 cm wide, and 45 cm high. In two frequencies at 7:00 a.m. and 4:00 p.m., feed was supplied ad libitum, tubular galvanized sheet feeders were used, and availability adjustments were made based on the difference between supply and rejection according to Iser et al. [16]. Moreover, automatic nipple drinkers were used to offer *ad libitum* water to the rabbits.

### 2.3. Growth Performance

During the study, body weight (BW) was determined at the beginning (35 days) and at the end (95 days) of the experiment using an OSBORNE^®^ (model 37473^®^, Kansas, MI, USA) digital scale with a precision of ±0.1 g. Overall mortality was calculated by comparing the number of dead animals to those that started the study. The feed supplied and rejected for a period of 24 h was used to calculate the daily feed intake (FI). The initial body weight (BW), final body weight, and the experimental days were used to determine the average daily gain (ADG). Likewise, the feed conversion ratio (FCR) was recorded as the kg of feed consumed to gain 1 kg of BW.

### 2.4. Intestinal Integrity

At 95 days old, 10 rabbits were randomly selected for each treatment. An electrical stun was performed with two electrodes under the following specifications: electricity 125 V, power 160 mA, 0.3 amps, frequency 50 Hz, and duration 2–3 s. After unconsciousness was verified, they were slaughtered, and six samples of intestinal tissue were taken [21].

After the tissues were washed, representative sections (1 cm^2^ samples) were taken from the middle longitudinal part of the duodenum (the area that corresponds to the junction with the stomach and up to the first intestinal loop) and from the middle longitudinal part of the cecum. Note that one section per intestinal segment (duodenum and cecum) was evaluated. Next, they were placed in 10% formalin in phosphate-buffered saline (pH 7.4) at 4 °C, which was subjected to the histological technique [8], dehydrated in a graduated series of ethanol, and embedded in paraffin. Subsequently, thick samples of 4 to 5 microns were taken with a longitudinal orientation to ensure that the complete layers of the intestine were present.

Next, in the tissues, a routine of deparaffinization, hydration, and staining (hematoxylin-eosin) was followed. Five slides per animal were evaluated, with five serial cuts per slide and five random fields per slide. The thickness of the muscular and mucosal layers in the duodenum and cecum was calculated. Furthermore, in the duodenum, the height, width, and number of villi were quantified. In the cecum, the depth, thickness, and number of the crypts were determined. An Axiostar microscope (Carl Zeiss, Oberkochen, Germany) was connected to a computer with Opti-AnalySIS Basic software, and 500× and 100× images were used to determine all study measurements. Finally, the ratio of villus height/depth of the crypt was calculated [8].

### 2.5. Statistical Analysis

The experimental study considered a completely randomized design. To determine the data normality and the variance uniformity, the Kolmogorov–Smirnov and Bartlett tests were used. Next, the data were processed by a simple classification analysis of variance (ANOVA). Where necessary, a post hoc analysis (Duncan) was used. Likewise, the Kruskal–Wallis test was performed to determine the number of villi and crypts. All analyzes were performed according to the statistical software SPSS version 23.0.

## 3. Results

Table 2 shows that the dietary use of the *Agave tequilana* stem powder did not statistically change the viability and FCR (*p* > 0.05); however, T3 increased BW, FI, and the ADG compared to a basal diet (T0), T1, and T2 during the 60 experimental days (*p* < 0.05).

Changes in the intestinal integrity of rabbits (95 days old) when using increasing levels of *A. tequilana* stem powder are shown in Table 3. T3 significantly (*p* < 0.05) enhanced the thickness of the muscular and mucous layers in the duodenum and cecum. Similar results were found for the height and villi thickness in the duodenum compared to T0, T1, and T2 (*p* < 0.05). However, these last treatments (T0–T2) showed (*p* < 0.05) the highest values for the depth zone of the crypts, without differences (*p* > 0.05) for the crypts thickness in the duodenum. Furthermore, T3 decreased (*p* < 0.05) the depth and crypts thickness in the cecum significantly in relation to the other treatments.

Figure 1 shows that T3 increased (*p* < 0.05) the villi number in the duodenum, although without notable differences among treatments (*p* > 0.05) for the crypts number in both intestinal portions (duodenum and cecum).

## 4. Discussion

The fact that supplementation of up to 1.5% of *Agave tequilana* stem powder did not change the viability of rabbits (Table 2) confirms that this natural product does not have toxic elements and/or harmful secondary metabolites. Similar results in mortality (6.25%) were found by Iser et al. [8] and Iser et al. [16] when they used up to 1.5% of *A. tequilana* and *A. fourcroydes* stem powder as a natural growth promoter. Likewise, Chávez et al. [14] and Sánchez et al. [15] informed us about a high viability in poultry and pigs when they carried out dietary experiments with dried-stem powder or *Agave tequilana* extract, thus demonstrating the safety of the tested natural product.

On the other hand, in laboratory animals, *Agave tequilana* fructans have been found to modulate the intestinal microbiota, the immune response, the anti-inflammatory activity, and the circulation of harmful serum lipids, which causes a maintenance of or reduction in body weight [22,23]. However, in the case of using rabbits as laboratory animals, T3 increased their body weight by 2.87% and their ADG by 4.37% when compared to the control group (Table 2). The growth-promoting effect of these products rich in fructans will apparently depend on the amount of this carbohydrate in the diet and the category and species of the animal under experimentation. Moreover, Santos-Zea et al. [24] have informed us that the *Agave tequila* stem is rich in secondary metabolites such as polyphenols and saponins, which are considered anti-nutritional factors. However, in adequate concentrations, these chemical compounds have antioxidant and anti-inflammatory effects that decrease low density lipoprotein (LDL) oxidation and regulate cell growth [25]. Apparently, the combined effect of fructans and the beneficial secondary metabolites had a positive effect on rabbits, although more experiments are necessary to verify this approach.

Table 2 also demonstrates that dietary use with dried-stem powder of *A. tequilana* (T3) provoked a higher feed intake (3.59%) and no change in feed conversion, which promoted a higher body weight, suggesting that perhaps favorable conditions in the gastrointestinal tract (GIT) demanded a higher intake of nutrients and energy (Table 3). Previous studies with agave stem (*tequilana* and *fourcroydes*) reported a higher feed intake associated with an increased weight gain in rabbits [8,16]. *Agave tequilana* stem powder also promoted feed intake due to the moderately sweet taste of this natural product as a result of the high concentration of fructans and fructose [26]. Furthermore, other studies using prebiotic compounds in rabbits are contradictory; Mourão et al. [27] and Bovera et al. [28] found that the dietary use with mannanoligosaccharides as a growth-promoter in rabbits increased feed intake, but Alvarado-Loza et al. [29] reported lower intake when they used insulin as part of the rabbit diet. The results (Table 2) demonstrate that dietary supplementation up to 1.5% of *Agave tequilana* stem powder has a natural growth-promoting effect in rabbits. This was supported by other research using *Agave fourcroydes* dried-stem as a prebiotic additive in broiler rabbits [8].

Additionally, *Agave tequilana* stem supplementation (1.5%) changed the thickness of the muscular and mucous layers in broiler rabbits; these intestinal layers are the most significant in digestive morphophysiology and in the diagnosis of digestive diseases; the first contributes to peristalsis and the movement of the feed chyme, which directs it along and out of the intestines, and the second produces mucus, lubricates the passage of the food chyme, and protects the GIT from the action of digestive enzymes [30]. Thus, an increase in the thickness of these intestinal layers (muscle and mucosa) with T3 is associated with higher intestinal health [6]. These results may be associated with a decrease in intestinal pH due to the increase in the population of cecal lactic acid bacteria (BAL), already confirmed by Martínez et al. [31] when using *Agave fourcroydes* stem powder (with similar chemical characteristics to *Agave tequilana*) up to 1.5% in rabbit diets. In this sense, Revolledo et al. [32] have informed us that better intestinal health, due to decreased adherence of pathogenic bacteria and intestinal damage due to competitive exclusion of bacteria, has a direct effect on the concentric layers thickness.

Furthermore, de Blas et al. [33] found a reduction in the thickness of the mucous layer due to the greater presence of *Helicobacter* spp., *Campylobacter* spp. and *C. perfringens*, which increased the mortality and decreased the productive response of rabbits. This intestinal layer is related to tissues involved in the defense of the host against infectious diseases. This allows for better immune activity and modulation of the intestinal barrier [32], which directly affect the productive performance of animals [3].

On the other hand, Raj et al. [34] reported that there is a close relationship between the thickness of the intestinal mucosa and the intestinal barrier. These authors reported that a significant decrease in this intestinal layer can decrease intestinal permeability due to the unlimited access of toxins, microorganisms, substances, macromolecules, and chemicals, which could cause gastrointestinal disturbances that directly affect animal productivity. It should be noted that the mucous layer thickness is greater in the duodenum than in the cecum due to absorptive activity in this portion. However, the muscle layer thickness is more significant due to the transport of feces, especially in rabbits due to cecotrophy [20].

Additionally, some authors [35] have suggested that the histo-morphometry of the villi and crypts are directly related to intestinal health and animal response. Thus, these results showed that the better conditions of the intestinal environment, due to a higher proliferation of lactic acid bacteria (LAB) [31] provided by T3, led to an increase (*p* < 0.05) in the height and thickness of the villi, and in the thickness of the intestinal mucosa (Table 3), thus denoting a more developed intestinal tissue. Other experiments with productive animals using high-fructan diets have reported similar responses in villus height and width [30,31]. The authors associated these results with greater health and intestinal absorption area. However, the studies by Mourão et al. [27], when they used diets with fructooligosaccharides in rabbits, did not find notable changes (*p* > 0.05) for these indicators.

It has been verified that the use of nutraceutical foods intervenes in the development of the GIT, especially in “villi height and crypts depth,” as well as in productive efficiency, nutrient absorption capacity, and reduction of metabolic requirements [35]. In this sense, the relationship of the villi height/crypts depth is used as a nutritional and health indicator to estimate the digestive processes and the absorption of nutrients in the gut [36]. Thus, the greatest capacity for digestion and absorption occurs when this ratio increases [37,38]. According to Vallejos et al. [30], a shortening of the intestinal villi in relation to a greater depth of the crypt causes a decrease in absorption cells and more secretory cells. On the other hand, the highest supplementation with *Agave tequilana* stem powder (1.5%) decreased the thickness and depth of the crypts in the duodenum (Table 3). According to Cai et al. [6], in apparently healthy animals, a greater crypt depth is related to the migration of specialized cells towards the villi, influenced by the shortening and decreased functionality of these structures due to multifactorial causes, although with greater emphasis due to the microbial dysbiosis. Likewise, other studies on the dietary use of up to 1.5% *Agave fourcroydes* stem powder compared to the control treatment showed a higher ratio of these intestinal structures [8]. It is noteworthy that the intestine has a rapid epithelial renewal, due to the shortening of the deep Lieberkühn crypts [33].

At the same time, in previous research, Iser et al. [8] reported that the dietary use of *Agave fourcroydes* stem powder (up to 1.5%) enhanced the number of intestinal villi, due to better gut health, which promoted the growth of rabbits. To our knowledge, this is the first study showing the effect of *Agave tequilana* powder on intestinal histo-morphometry of rabbits. Moreover, the number and size of the villi will depend on the number of cells that compose it. In this way, the functional integrity of the villi cells, both in the luminal membrane and in the base-lateral membrane, are directly related to the absorption of nutrients [39]. It is important to clarify that each villus corresponds to a crypt; the numerical difference in the duodenum is due to the cross section made to the tissues for this analysis [30].

Generally, gastrointestinal problems in human beings and animals provoke changes in the villi structure (mainly atrophy or shortening), especially due to the presence of pathogenic bacteria or other associated pathologies [38]. As mentioned above, in these apparently healthy animals (rabbits), supplementation with *Agave tequilana* powder (1.5%) increased the villi thickness and decreased the crypts depth, which provoked a reduction in the inter-villus space, with a greater quantification of these filaments (villi) at a field reading with 100× and 500× under the microscope. In this sense, Gāliņa et al. [40] found that various nutraceuticals increased the number of intestinal villi in an apparently healthy animal model. In a similar vein, Mourao et al. [28] suggested that the diets with mannan-oligosaccharides provoked a greater height of intestinal villi, associated with the growth of beneficial cecal bacteria in rabbits. This shows that the number of functional villi in the duodenum changes due to the effect of the diet supplied, considering that dietary treatments did not cause enteric problems in these rabbits.

## 5. Conclusions

The results indicate that the dietary supplementation of 1.5% of *A. tequilana* (Weber var. Azul) stem powder in broiler rabbits (35–95 days old) improved the growth performance and histomorphometry of the concentric layers (muscle and mucosa), villi, and crypts as indicators of intestinal health, which justified the natural growth-promoting effect of this product in rabbit production.

## Figures and Tables

**Figure 1 animals-12-01117-f001:**
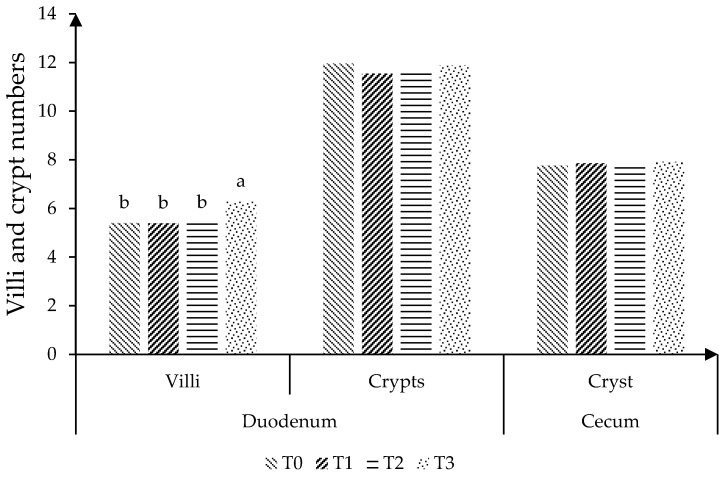
Effect of *A. tequilana* stem powder on the number of villi (SEM ± 0.132; *p*-value 0.001) and crypts in the duodenum (SEM ± 0.494; *p*-value 0.088) and on the crypts number (SEM ± 0.259; *p*-value 0.664) in the cecum of rabbits. ^a,b^ Means with different letters among treatments differ at *p* < 0.05.

**Table 1 animals-12-01117-t001:** Ingredients and nutritional contributions of the diet for broiler rabbits (35 to 95 days old).

Ingredients	Basal Diet (%)
Wheat straw	17.4
Alfalfa hay	12.0
Barleycorn	19.0
Wheat bran	24.0
Sunflower meal	12.0
Soymeal	11.0
Soy oil	2.88
Sodium chlorine	0.50
Monocalcium phosphate	0.50
L-lysine	0.09
L-threonine	0.08
DL-methionine	0.05
Premix ^1^	0.50
Calculated nutritional contributions	
Crude protein (%)	16.70
Digestible energy (MJ/kg)	9.92
Neutral detergent fiber (%)	35.78
Detergent acid fiber (%)	19.21
Lysine (%)	0.77
Methionine + cystine (%)	0.59
Threonine (%)	0.65
Ashes (%)	5.37

^1^ Each kg contains: vitamin A 12,000 IU, vitamin D3 2000 IU, vitamin B2 4160 IU, Niacin 16,700 IU, pantothenic acid 8200 IU, vitamin B6 3420 IU, folic acid 0.980 g, vitamin B12 16 mg, vitamin K 1560 IU, Vitamin E 16 g, BHT 8.5 g, cobalt 0.750 g, copper 3.5 g, iron 9.86 g, manganese 6.52 g, sodium 0.870 g, zinc 4.24 g, and selenium 6.67 g.

**Table 2 animals-12-01117-t002:** Effect of *A. tequilana* stem powder on performance of rabbits (35–95 days old).

	Treatments					
Items	T0	T1	T2	T3	SEM±	*p*-Value
Viability (%)	95.00	95.00	95.00	95.00		
Initial body weight	774.89	767.75	768.5	771.66	4.430	0.890
Final body weight (g)	2478.89 ^b^	2472.35 ^b^	2468.90 ^b^	2550.06 ^a^	14.728	0.041
Feed intake (g/rabbit/day)	120.49 ^b^	119.60 ^b^	119.50 ^b^	124.81 ^a^	0.437	<0.001
Average daily gain (g/rabbit/day)	28.40 ^b^	28.41 ^b^	28.34 ^b^	29.64 ^a^	0.296	0.049
Feed conversion ratio	4.24	4.21	4.22	4.21	0.034	0.059

^a,b^ Means with different letters in the same row differ at *p* < 0.05. T0: basal diet; T1: 0.5% of *Agave tequilana* stem powder. T2: 1.0% of *Agave tequilana* stem powder. T3: 1.5% of *Agave tequilana* stem powder.

**Table 3 animals-12-01117-t003:** Effect of *A. tequilana* stem powder on intestinal integrity of rabbits (95 days old).

	Treatment					
Morphometry (µm)	T0	T1	T2	T3	SEM±	*p*-Value
Duodenum						
Muscle thickness	119.20 ^b^	121.72 ^b^	119.88 ^b^	150.00 ^a^	6.786	<0.001
Mucous thickness	1063.24 ^b^	1056.56 ^b^	1023.84 ^b^	1199.9 ^a^	31.800	<0.001
Villi height	891.70 ^b^	894.12 ^b^	892.22 ^b^	1027.96 ^a^	25.041	<0.001
Villi thickness	109.00 ^b^	108.03 ^b^	110.52 ^b^	149.80 ^a^	6.500	<0.001
Crypt area	321.21 ^a^	322.15 ^a^	322.24 ^a^	267.32 ^b^	7.589	<0.001
Crypt depth	99.55 ^a^	98.65 ^a^	98.20 ^a^	72.68 ^b^	7.215	<0.001
Crypt thickness	66.89	65.76	65.77	65.76	4.122	0.314
Villi/Crypts *	8.95 ^b^	9.06 ^b^	9.06 ^b^	14.14 ^a^	1.251	<0.001
Cecum						
Muscle thickness	286.79 ^b^	284.00 ^b^	287.32 ^b^	396.04 ^a^	28.069	<0.001
Mucosa thickness	425.99 ^b^	426.50 ^b^	428.20 ^b^	441.00 ^a^	3.482	<0.001
Crypt depth	244.30 ^a^	243.26 ^a^	245.76 ^a^	228.76 ^b^	4.923	<0.001
Crypt thickness	121.36 ^a^	121.28 ^a^	121.75 ^a^	97.84 ^b^	5.730	<0.001

^a,b^ Means with different letters in the same row differ at *p* < 0.05. * Villus height/crypt depth. T0: basal diet; T1: 0.5% of *Agave tequilana* stem powder. T2: 1.0% of *Agave tequilana* stem powder. T3: 1.5% of *Agave tequilana* stem powder.

## Data Availability

The data presented in this study are available on request from the corresponding author.

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
