# Peer review of "Dietary Supplementation with *Agave tequilana* (Weber Var. Blue) Stem Powder Improves the Performance and Intestinal Integrity of Broiler Rabbits"

_animals, 2022, doi:10.3390/ani12091117_

Round 1

Reviewer 1 Report

Manuscript review - Animals-1652015

Tittle: Dietary supplementation with Agave tequilana stem powder improves the performance and intestinal integrity of broiler rabbits

General Comments

This study documents the effect of increasing doses of Agave tequilana stem powder on the performance and intestinal morphology of growing rabbits. The data is consistent, sample size is adequate, but several issues must be addressed. First, the description of tissue sample analysis is missing. A detailed explanation on how this was carried out is essential. Although diet composition was described in a previous paper, it must be presented here too. And chemical analyses must be presented as well. There was a dilution effect of the diet due to the increasing levels of Agave tequilana stem powder added. That was not considered. English language is poor and must be improved (lack of punctuation, misuse of prepositions, defective verb conjugation, unusual vocabulary). Another drawback is a repetitive discussion.

Specific comments

Simple summary

Line 15: Delete “the use of”

Line 16: change to “these synthetic products have been used to a lesser extent than for other monogastrics”

Line 18: The expression “rich in chemical benefits” does not seem right. Please be more specific. What are you referring to? Chemical compounds with beneficial effects on animal health?

Line 19: “secondary metabolites benefits responsible for “. Again, the word “benefits” was misused, it does not make sense.

Abstract

Lines 28-29: “Oral administration” is not appropriate. It gives the impression that you gave it separate from the diet. Please rewrite.

The use of acronyms (T0, T1, T2…) is confusing for the readers, and sometimes even for the authors! For instance, in Figure 1 the authors indicated “T4” (typo?) that does not exist! I recommend avoiding these acronyms throughout the manuscript and using the actual product levels instead.  

Line 30: “productivity” is vague. What do you mean? Body weight? Weight gain?

Line 32: change to “and number of villi”. But you still have to say in which portion of the intestine.

Lines 32-33: I believe you broke the logics here. Please continue to compare the highest level to the others, saying that it was associated with the lowest crypt area and height. And the same for the cecum (lines 34-35)

Lines 33 and 34: “provoked” is too strong and not necessary. Simply say that such treatment increased the villus height:crypt depth relationship.

Introduction

Lines 57-59: this sentence does not make sense due to lack of punctuation. Please correct.

Lines 69- 71: change “Animal studies of zootechnical interest…. respectively” to “Studies on farm animals reported that Agave tequilana stem powder acted as a natural growth promoter in diets increasing the population of beneficial cecal bacteria and modifying harmful serum lipids in pigs and poultry, respectively”.

Line 73: delete “also”.

Line 74: change “unchanged” to “without changing”

Lines 74-78: “However, no studies…. rabbit production”. This sentence is too long and confusing. Please rephrase.

Lines 78-81: When you refer to the structures you have to specify the site: small intestine, cecum, etc.

Material and methods

Lines 86-88: There is no information about the animal housing (except for cages) and how did you maintain ambient temperature and relative humidity constant. Was it a closed air-conditioned building? Or an opened building? When you say that temperature and relative humidity were “controlled” by a hygro-thermometer, you actually meant that they were “measured” (not controlled!) daily using this equipment.

Line 92: You forgot to say (it was mentioned in the Abstract) that the rabbits were housed two per cage (replicate).

Lines 93-94: I suggest avoiding these acronyms (T0, T1, etc). Please present the results of a chemical analysis of the Agave tequilana stem powder.

Lines 97-99: Even though the same basal diet was used in a previous experiment, the information about diet composition is essential. Please include it in this manuscript along with the results of the chemical analysis.

There is a dilution effect of the diet as you add increasing levels of Agave tequilana stem powder. Was this taken into consideration? It might be small, but it exists. You could have used an inert product (clay) or fibrous feed in the basal diet and replace it with Agave tequilana stem powder in the treatment diets. You should, at least say, if it was the case, that this effect was not considered and why.

According to the Table of diet composition in reference number 17 (Iser et al., 2019), the NDF level in your basal diet is suboptimal. The levels recommended for rabbits by De Blas & Mateos (2010) vary from 32 to 35%. Therefore, could the beneficial effect of Agave tequilana stem powder be attributed to increasing NDF in the diet, reaching a level closer to the recommendations for rabbits?

Line 102: change to “animal species”

Line 105: change to “feed was supplied”

Lines 110-111: this sentence does not make any sense. The rabbits were fasted? What for? Or weighed? Please clarify.

Line 118: Ten rabbits per treatment were euthanized. What was the method used for euthanasia? How were these rabbits chosen? At random?

Lines 118- 131: Precise and detailed information must be presented on the portions of the duodenum and cecum that were sampled. Additionally, how many structures were measured per sample? At what frequency? What distances were measured in each type of structure? All these information must be presented in detail to validate your data. Otherwise, they are useless. Ghanima et al., (2020). PLoS ONE 15(6): e0234076 provides an example on how to describe these measurements in detail.

Results

Table 1: Please include BW at 35 days (initial weight).

Figure 1: there was no T4

Discussion

Line 167: It is the viability of rabbits in each treatment, not of the experiment.

Line 167-172: this sentence is too long and confusing. Please rephrase.

Line 179: Intestinal microbiota is preferred (over microflora)

Lines 179-184: caution should be exercised when comparing rabbits to laboratory animals. Mice are omnivorous whereas rabbits are herbivorous with highly specific adaptations to the vegetarian diet.

Lines 191-194: I am afraid that this increase in feed intake for the higher level of Agave tequilana stem powder might be due to a better balance of the diet mentioned above…

Lines 222-223: “increased mortality and productive response of rabbits”. What do you mean?

Line 238: What does “LAB” stand for?

Line 254: It would be much better to say that the higher level of Agave tequilana stem powder decreased thickness and depth of the 254 intestinal crypts (as I indicated in the Abstract)

Lines 260-263: There is no need to repeat the findings of Pinheiro et al. here.

Lines 268-271: “Moreover….in an inflamed intestine”. This information is not relevant for your study. Delete this sentence.

Author Response

Reviewer 1

Dear reviewer,

Thank you very much for your comments on our manuscript.

Reviewer: This study documents the effect of increasing doses of Agave tequilana stem powder on the performance and intestinal morphology of growing rabbits. The data is consistent, sample size is adequate, but several issues must be addressed. First, the description of tissue sample analysis is missing. A detailed explanation on how this was carried out is essential. Although diet composition was described in a previous paper, it must be presented here too. And chemical analyses must be presented as well. There was a dilution effect of the diet due to the increasing levels of Agave tequilana stem powder added. That was not considered. English language is poor and must be improved (lack of punctuation, misuse of prepositions, defective verb conjugation, unusual vocabulary). Another drawback is a repetitive discussion.

Authors: The manuscript has been reviewed by a native English-speaking researcher. In addition, we have considered most of your suggestions.

Reviewer: Line 15: Delete “the use of”

Authors: Done

Reviewer: Line 16: change to “these synthetic products have been used to a lesser extent than for other monogastrics”

Authors: Done

Reviewer: Line 18: The expression “rich in chemical benefits” does not seem right. Please be more specific. What are you referring to? Chemical compounds with beneficial effects on animal health?

Line 19: “secondary metabolites benefits responsible for “. Again, the word “benefits” was misused, it does not make sense.

Authors: We improved the wording for a better understanding.

Reviewer: Lines 28-29: “Oral administration” is not appropriate. It gives the impression that you gave it separate from the diet. Please rewrite.

Authors: Done

Reviewer: The use of acronyms (T0, T1, T2…) is confusing for the readers, and sometimes even for the authors! For instance, in Figure 1 the authors indicated “T4” (typo?) that does not exist! I recommend avoiding these acronyms throughout the manuscript and using the actual product levels instead.  

Authors: The authors consider that the use of acronyms is common in current scientific writing because it helps understanding and avoids repetition. We show some examples of our articles published in "Animals" that use the acronyms.

Martínez, Y., Almendares, C., Hernández, C., Avellaneda, M. C., Urquía, M. N. & Valdivié, M. 2021. Effect of acetic acid and sodium bicarbonate supplemented to drink water on water quality, growth performance, organ weights, cecal traits and hematological parameters of young broilers. Animals. 11(7): 1865. https://doi.org/10.3390/ani11071865.

Martínez, Y., Altamirano, E., Ortega, V., Paz, P. & Valdivié, M. 2021. Effect of age on the immune and visceral organ weights and cecal traits in modern broilers. Animals. 11: 845. https://doi.org/10.3390/ani11030845.

Betancur, C., Martínez, Y.*, Merino-Guzman, R., Hernandez-Velasco, X., Castillo, R., Rodríguez, R. & Téllez, G. 2020. Evaluation of oral administration of Lactobacillus plantarum CAM6 strain as an alternative to antibiotics in weaned pigs. Animals. 10(1218). http://doi.org/10.3390/ani10071218. *Corresponding author.

Reviewer: Line 30: “productivity” is vague. What do you mean? Body weight? Weight gain?

Authors: Done. Body weight

Reviewer: Line 32: change to “and number of villi”. But you still have to say in which portion of the intestine.

Authors: Done. The study was in the duodenum

Reviewer: Lines 32-33: I believe you broke the logics here. Please continue to compare the highest level to the others, saying that it was associated with the lowest crypt area and height. And the same for the cecum (lines 34-35)

Lines 33 and 34: “provoked” is too strong and not necessary. Simply say that such treatment increased the villus height:crypt depth relationship.

Authors: Done. We improved the wording for a better understanding.

Reviewer: Lines 57-59: this sentence does not make sense due to lack of punctuation. Please correct.

Authors: Done. We improved the wording for a better understanding.

Reviewer: Lines 69- 71: change “Animal studies of zootechnical interest…. respectively” to “Studies on farm animals reported that Agave tequilana stem powder acted as a natural growth promoter in diets increasing the population of beneficial cecal bacteria and modifying harmful serum lipids in pigs and poultry, respectively”.

Line 73: delete “also”.

Line 74: change “unchanged” to “without changing”

Authors: Done. We improved the wording for a better understanding.

Reviewer: Lines 74-78: “However, no studies…. rabbit production”. This sentence is too long and confusing. Please rephrase.

Authors: Done. We improved the wording for a better understanding.

Reviewer: Lines 78-81: When you refer to the structures you have to specify the site: small intestine, cecum, etc.

Authors: Done

Reviewer: Lines 86-88: There is no information about the animal housing (except for cages) and how did you maintain ambient temperature and relative humidity constant. Was it a closed air-conditioned building? Or an opened building? When you say that temperature and relative humidity were “controlled” by a hygro-thermometer, you actually meant that they were “measured” (not controlled!) daily using this equipment.

Authors: Done. We improved the wording for a better understanding.

Line 92: You forgot to say (it was mentioned in the Abstract) that the rabbits were housed two per cage (replicate).

Authors: Done.

Reviewer: Lines 93-94: I suggest avoiding these acronyms (T0, T1, etc). Please present the results of a chemical analysis of the Agave tequilana stem powder.

Authors: Done.

Reviewer: Lines 97-99: Even though the same basal diet was used in a previous experiment, the information about diet composition is essential. Please include it in this manuscript along with the results of the chemical analysis.

Authors: Done.

Reviewer: There is a dilution effect of the diet as you add increasing levels of Agave tequilana stem powder. Was this taken into consideration? It might be small, but it exists. You could have used an inert product (clay) or fibrous feed in the basal diet and replace it with Agave tequilana stem powder in the treatment diets. You should, at least say, if it was the case, that this effect was not considered and why.

Authors: As in other experimental works, Agave tequilana stem powder was considered as a functional additive, the levels of protein and fiber, for example, are low for rabbits, its greater effect in its prebiotic action due to the high concentration of inulin-type fructans, thus, this positive effect was demonstrated in intestinal health indicators, for the first time reported to the international scientific community. However, for future work we will consider your recommendations.  

Reviewer: According to the Table of diet composition in reference number 17 (Iser et al., 2019), the NDF level in your basal diet is suboptimal. The levels recommended for rabbits by De Blas & Mateos (2010) vary from 32 to 35%. Therefore, could the beneficial effect of Agave tequilana stem powder be attributed to increasing NDF in the diet, reaching a level closer to the recommendations for rabbits?

Authors: You are right. We correct the calculated nutritional contributions, and the contributions are as follows: neutral fiber detergent: 35.78% and acid fiber detergent: 19.21%. The table below shows the levels of NDF and ADF used in the diets. Thank you for this appreciation.

Ingredients

NDF (%)

ADF (%)

Wheat straw

72

46.4

Alfalfa hay

38

28.6

Barleycorn

18.1

5.5

Wheat bran

40.3

13.4

Sunflower meal

34.8

22.1

Soymeal

12.8

7.2

Reviewer: Line 102: change to “animal species”

Line 105: change to “feed was supplied”

Authors: Done

Reviewer: Lines 110-111: this sentence does not make any sense. The rabbits were fasted? What for? Or weighed? Please clarify.

Authors: Done. We improved the wording for a better understanding.

Reviewer: Line 118: Ten rabbits per treatment were euthanized. What was the method used for euthanasia? How were these rabbits chosen? At random?

Authors: Done. We improved the wording for a better understanding.

Reviewer: Lines 118- 131: Precise and detailed information must be presented on the portions of the duodenum and cecum that were sampled. Additionally, how many structures were measured per sample? At what frequency? What distances were measured in each type of structure? All these information must be presented in detail to validate your data. Otherwise, they are useless. Ghanima et al., (2020). PLoS ONE 15(6): e0234076 provides an example on how to describe these measurements in detail.

Authors: Done. We improved the wording for a better understanding.

Reviewer: Figure 1: there was no T4

Authors: Done.

Reviewer: Line 167: It is the viability of rabbits in each treatment, not of the experiment.

Authors: We improved the wording for a better understanding.

Reviewer: Line 167-172: this sentence is too long and confusing. Please rephrase.

Authors: We improved the wording for a better understanding.

Reviewer: Line 179: Intestinal microbiota is preferred (over microflora)

Authors: Done

Reviewer: Lines 179-184: caution should be exercised when comparing rabbits to laboratory animals. Mice are omnivorous whereas rabbits are herbivorous with highly specific adaptations to the vegetarian diet.

Authors: The comparison with laboratory animals of the use of prebiotics is due to the fact that rabbits are one of the most used laboratory animals for these purposes, and clearly their condition as herbivorous monogastric animals can influence the productive response, as was the case. . “the growth-promoting effect of these products rich in fructans, will depend on the amount of this carbohydrate in the diet and the category and animal species under experimentation”. in the second paragraph, we eliminate those referring to the consumption of laboratory animals.

Reviewer: Lines 191-194: I am afraid that this increase in feed intake for the higher level of Agave tequilana stem powder might be due to a better balance of the diet mentioned above…

Authors: This question was resolved previously, with the incorporation of the actual contribution of neutral detergent fiber and acid detergent fiber.

Reviewer: Lines 222-223: “increased mortality and productive response of rabbits”. What do you mean?

Authors: We improved the wording for a better understanding.

Reviewer: Line 238: What does “LAB” stand for?

Authors: Lactic acid bacteria

Reviewer: Line 254: It would be much better to say that the higher level of Agave tequilana stem powder decreased thickness and depth of the 254 intestinal crypts (as I indicated in the Abstract)

Authors: Done.

Reviewer: Lines 260-263: There is no need to repeat the findings of Pinheiro et al. here.

Authors: OK

Reviewer: Lines 268-271: “Moreover….in an inflamed intestine”. This information is not relevant for your study. Delete this sentence.

Authors: Done.

Reviewer 2 Report

Interesting study, should be published after minor revision.

Please find my corrections and comments in the attached file.

Author Response

Reviewer 2

Dear reviewer,

Thank you very much for your comments on our manuscript.

We have corrected the manuscript according to your suggestions.

Reviewer: you can describe that there was a significant decrease in villi/crypt ratio.

Authors:  Done.

Reviewer: What do you mean exactly with "viability of the experiment (95%)"? The viability of the rabbits? Survival rate? The feasibility of the experiment? Anything else?

Authors: We changed the wording for a better understanding

Reviewer 3 Report

The authors present a manuscript evaluating positive effects of Agave plant feeding to rabbits. English grammar seems ok. The figures and tables are clear and easy to understand. The study showed that the proposed feed increased muscle significantly and had a minor but positive effect in weight gain. The article seems scientifically sound, methodology wise, and the statistical work seems coherent. The number of subjects was adequate. The article seemed interesting and it showed promising results.

A minor suggestion: Clearly state the significance level of 0.05 and the null hypothesis of equal averages in the statistical section

Author Response

Reviewer 3

Dear reviewer,

Thank you very much for your comments on our manuscript.

 Reviewer: A minor suggestion: Clearly state the significance level of 0.05 and the null hypothesis of equal averages in the statistical section

Authors: Done

Round 2

Reviewer 1 Report

Most of my suggestions were considered and changes were made to the manuscript. 

However, some critical information was not yet provided. I will try to be more specific about what is missing. In order to assess intestinal integrity, villi number, height and width in the duodenum and thickness of mucosal layers in the duodenum and cecum were determined. To make these measurements:

_How many sections from each intestinal segment were evaluated?

-How many villi were inspected in each section?

-How many values were obtained from each sample? 

You have to provide these essential data.

Author Response

Dear reviewer,
We have incorporated all your questions into materials and methods.
Thanks for the support.